# Comparison of different sequencing techniques for identification of SARS-CoV-2 variants of concern with multiplex real-time PCR

Diyanath Ranasinghe[1], Tibutius Thanesh Pramanayagam Jayadas[1], Deshni Jayathilaka[1], Chandima Jeewandara[1], Osanda Dissanayake[1], Dinuka Guruge[2], Dinuka Ariyaratne[1], Dumni Gunasinghe[1], Laksiri Gomes[1], Ayesha Wijesinghe[1], Ruwan Wijayamuni[1], Gathsaurie Neelika Malavige[1]*

1 Allergy Immunology and Cell Biology Unit, Department of Immunology and Molecular Medicine, University of Sri Jayewardenepura, Nugegoda, Sri Lanka, 2 Colombo Municipal Council, Colombo, Sri Lanka

⊕ These authors contributed equally to this work.
* gathsaurie.malavige@ndm.ox.ac.uk

**Data Availability Statement:** All relevant data are within the paper and its Supporting Information files.

## Abstract

As different SARS-CoV-2 variants emerge and with the continuous evolvement of sub lineages of the delta variant, it is crucial that all countries carry out sequencing of at least >1% of their infections, in order to detect emergence of variants with higher transmissibility and with ability to evade immunity. However, due to limited resources as many resource poor countries are unable to sequence adequate number of viruses, we compared to usefulness of a two-step commercially available multiplex real-time PCR assay to detect important single nucleotide polymorphisms (SNPs) associated with the variants and compared the sensitivity, accuracy and cost effectiveness of the Illumina sequencing platform and the Oxford Nanopore Technologies' (ONT) platform. 138/143 (96.5%) identified as the alpha and 36/39 (92.3%) samples identified as the delta variants due to the presence of lineage defining SNPs by the multiplex real time PCR, were assigned to the same lineage by either of the two sequencing platforms. 34/37 of the samples sequenced by ONT had <5% ambiguous bases, while 21/37 samples sequenced using Illumina generated <5%. However, the mean PHRED scores averaged at 32.35 by Illumina reads but 10.78 in ONT. This difference results in a base error probability of 1 in 10 by the ONT and 1 in 1000 for Illumina sequencing platform. Sub-consensus single nucleotide variations (SNV) are highly correlated between both platforms ($R^2 = 0.79$) while indels appear to have a weaker correlation ($R^2 = 0.13$). Although the ONT had a slightly higher error rate compared to the Illumina technology, it achieved higher coverage with a lower number or reads, generated less ambiguous bases and was significantly less expensive than Illumina sequencing technology.

**Funding:** GNM, CJ: World Health Organization GNM, CJ: World bank, Sri Lanka Covid 19 Emergency Response and Health Systems Preparedness Project (ERHSP) of Ministry of Health Sri Lanka funded by World Bank.

**Competing interests:** The authors have declared that no competing interests exist.

## Introduction

Since the emergence of the SARS-CoV-2 virus two years ago, it continues to evolve giving rise to many variants with higher transmissibility and immune evasion abilities [1]. Despite a lower mutation rate of SARS-CoV-2 compared to many other RNA viruses such as HIV and influenza, [2], many SARS-CoV-2 variants of concern (VOCs) have emerged, which continue to drive the pandemic. Although the ongoing COVID-19 vaccine drive and other control measures has resulted in a reduction in the number of cases and deaths, many countries still report intense transmission rates [3]. This high rate of continued transmission of the virus is likely to give rise to new variants, which may evade immunity induced by vaccines, or may have a higher transmissibility than the dominant delta variant. This is already evident with the emergence of many delta-sub lineages [4], which some are under investigation for higher transmissibility. In order to detect such possible new variants emerging, the WHO has recommended that all countries carry out genomic sequencing in at least 1% of their infections [5].

Although there are many sub-lineages of delta emerging [4], so far the WHO has named four SARS CoV-2 variants as variants of concern (VOC), which are they are alpha (B.1.1.7), beta (B.1.351), gamma (P.1) and delta (B.1.617.2)[5]. Even though sequencing is the gold standard to identify VOCs, it is time consuming and expensive for many lower income and lower middle-income countries. In addition, although many developed countries sequence over 5% of their positive cases, many lower income countries are unable to sequence 1% and many sequence <0.2% [6]. Therefore, in order to carry out surveillance to monitor the current VOCs and to identify mutations of concern than may occur in such variants, cheaper and rapid methods are required that can be used in resource poor settings. The B.1.1.7 variant (alpha) was identified initially in the UK due to the changes in the SARS-CoV-2 real-time PCR results that occurred in the primers targeting the spike protein (S-drop) when using the Taq Path (Thermo Fisher Scientific Inc.) [7]. Subsequently, many multiplex real-time PCR assays were developed to identify Single Nucleotide Polymorphisms (SNPs) that associate with these four VOC such as the HV69-70del, K417N, N417T, W152C, E484K, N501Y, L452R, D614G, P681H or V1176F [8]. The combination of these different SNPs in the SARS-CoV-2 which can be identified by these multiplex real-time PCR can therefore be used to identify these four VOCs [9].

Many sequencing platforms are currently used to sequence the SARS-CoV-2 virus, which have their advantages and disadvantages [10]. Comparison of different sequencing protocols have shown that they have significant differences in sensitivity, reproducibility, and precision for detection of SNPs [11]. However, with the continued emergence and spread of new delta sub lineages, in addition to carrying out surveillance for VOCs, it is crucial to carry out whole genomic sequencing to identify variants which develop mutations of concern, that may evade immunity or may associate with higher transmissibility. However, as many lower income countries such as Sri Lanka, do not have adequate resources to carry out sequencing of >1% of the positive samples, we investigated the sensitivity and accuracy of a two-step commercially available multiple real-time PCR assay for detection of different SNPs associated with VOCs. In addition, we compared to sensitivity, reproducibility, precision of identification SNPs and the cost of sequencing using the Illumina and Oxford Nanopore sequencing platforms.

## Methodology

### Samples

Initial RT-PCR of diagnostic sputum or nasopharyngeal swabs were carried out on 190 samples using TaqPath COVID-19 CE-IVD RT-PCR kit (Thermo Fisher Scientific, USA)

according to the manufacturer's instructions. We have tested these samples during April to August 2021, which was the period Sri Lanka experienced its B.1.1.7 (Alpha) outbreak (mid-April) followed by a massive B.1.617.2 (Delta) outbreak in July 2021. Of the 190 samples, all analysed by the multiplex real-time PCR to detect SNPs of VOCs. 97 were subjected to sequencing by the Illumina platform, 56 by the Oxford nanopore sequencing technology and 37 by both methods. Those with a cycle threshold (Ct) values of <30 were subjected to genomic sequencing and further multiplex q RT-PCR for identification of SNPs associated with VOCs. Briefly, viral RNA was extracted using QIAamp viral RNA mini kit (Qiagen, USA), SpinStarTM Viral Nucleic Acid Extraction kit 1.0 (ADT Biotech, Malaysia) or FastGene RNA Viral Kit, (Nippon Genetics, Germany) according to manufacturer's instructions.

Ethical approval for the study was obtained by the Ethics Review Committee of the University of Sri Jayewardenepura. The samples for RT-PCR and sequencing of these individuals was sent to our laboratory for diagnostic purposes and for sequencing in order to carry out surveillance to detect the circulating SARS-CoV-2 variants. The data was fully anonymized before we accessed it for the analysis and the Ethics Review Committee provided a waiver for informed written consent in order for the details of these patients be included in this study.

## Multiplex qRT-PCR for identification of SNPs in VOCs

Multiplex qRT-PCR to detect SNPs was carried out using the SARS-COV-2 variant PCR Allplex™ (Seegene, South Korea) assay 1 and assay 2 in all 190 samples. The variant assay 1 detects SNPs for 501Y, E484K and 69–70 deletion, whereas variant assay 2 detects the SNPs L452R, W152C, K417T and K17N. The variant PCR assays were carried out according to the manufacturer's instructions using 5μl of the extracted RNA from samples (S1 methods). All the thermal cycling steps were carried out in a CFX96 Deep Well, Real time System (Bio-Rad, USA).

## Genomic sequencing using Oxford Nanopore (ONT) platform

The extracted RNA of 56/190 samples were subjected to nanopore sequencing according to the manufacture's instruction using the SQK-RBK110.96 rapid barcoding kit (ONT, Oxford, UK). 1200 bp tiled PCR amplicons were generated with midnight primers as described in Freed et al., 2020 [12]. All the thermal cycling steps were carried out in a QuantStudio™ 5 Real-Time PCR Instrument (Applied biosystems, Singapore). Barcodes were attached to resulting amplicons and pooled together before the clean-up step. Finally, 800ng of library was loaded into R9.4.1 flow cell and sequenced on the Oxford Nanopore Minion Mk1C. The run was stopped once desired number of reads (~15,000 reads per sample) were achieved.

## Genomic sequencing using the illumina platform

97 were subjected to sequencing by the Illumina sequencing platform. Library preparation Illumina sequencing was carried out using the AmpliSeq for Illumina SARS-CoV-2 Community Panel, in combination with AmpliSeq for Illumina library prep, index, and accessories (Illumina, San Diego, USA). The methodology is further described in (S1 methods).

## Bioinformatics and statistical analysis

Data from Illumina Nextseq 550 platform were base called using inbuilt bcl2fastq and the resulting data were analyzed using the Dragen somatic pipeline on Basespace sequencing hub [13]. Resulting Binary alignment files (BAM), variant call files (VCF) and consensus fasta sequences were taken further for comparison. ONT fast5 data were base called and demultiplexed using guppy version 4.4.2 occupying the fast model with a 60 single end barcode score.

Resultant fastq files were analyzed using Nextflow wf-artic version v0.3.2 (https://github.com/epi2me-labs/wf-artic) with "fast_min_variant_c507" model. Read length cutoff values obtained were in the range of 150bp and 1200bp, without barcode trimming. Resulting BAM, VCF and consensus fasta sequences were carried forward. Fastq and Bam statistics were generated using fastp 2 [14], Nanocomp 3 [15], samtools version 1.9 and in-house scripts. VCF data were processed using vcftools version 0.1.15 and bcftools version 1.7 [16]. Multi-way pileup (mpileup) was made from ONT and Illumina alignments before analyzing with VarScan2 [17] for detection of low frequency variants.

Fasta sequences were run through the Covsurver variant annotation pipeline (https://www.gisaid.org/epiflu-applications/covsurver-mutations-app/) and for lineage assessment each sequence was assessed using Pangolin version v3.1.11 [18] with the lineage version of 2021-08-09 and Serious constellations of reoccurring phylogenetically-independent origin (Scorpio) version 0.3.12 (https://github.com/cov-lineages/scorpio). All the consensus fasta sequences were aligned to the reference sequence MN908947.3 using MAFFT version 7.487 and a maximum likelihood phylogenetic tree was created using IQ-TREE version 1.6.1 [19] occupying the best fit TIM2+F+R2 model with 1000 bootstrap replicates. Raw fastq and consensus fasta files for 74 datasets were deposited to GISAID (S1 Table). Plots were generated in R version 4.0.1 using ggplot2, dplyr and tidyverse.

## Results

Multiplex real-time PCR to detect SNPs of VOCs was carried out initially in 190 deidentified samples. These 190 samples were then sequenced either on Illumina or ONT (depending on the availability of sequencing reagents) to evaluate the accuracy of the variant PCR. 37/190 of the samples were sequenced with both the Illumina and ONT platforms to compare the quality and accuracy of the data generated by these two platforms.

### Comparison between next generation sequencing with multiplex real-time PCR assays for detection of SNP (variant PCR assay)

143/190 samples were positive for the SNP N501Y and HV69/70 deletion in the spike protein by the multiplex PCR and therefore, considered to be as B.1.1.7. The samples in which these two SNPs were not detected (n = 47) were re-screened using the variant assay 2, which detected the SNPs L452R, W152C, K417T and K17N. Based on the results of the variant assay 2, 39 samples were positive only for the spike L452R mutation and were therefore, considered to be B.1.617. One sample with spike K417N mutation along with N501Y and HV69/70 deletion from the assay was considered to be B.1.351. 7/190 samples were not assigned to be any VOC as they did not have any of the SNPs associated with the VOCs.

105/109 (96.3%) samples interpreted as B.1.1.7 by the variant PCR assay were assigned to the same Pango lineage when sequenced with Illumina workflow, while the 4/109 samples had inadequate sequencing coverage to be assigned a lineage. 17/20 (85%) samples which were considered to be B.1.617 due to the presence of the spike L452R, were confirmed as B.1.617.2 by Illumina sequencing. Rest of the B.1.617 (3/20) were assigned to B.1.1.372, C.36 and B.1. 5/5 (100%) samples in which the SNPs were not detected, were classified into B.1, B.1.1 or B.1.411 with the Illumina sequencing workflow.

52/55 (94.5%) samples interpreted as B.1.1.7 were confirmed by the ONT sequencing workflow while 3/55 lacked sequencing coverage. 32/35 (91.4%) samples concluded to be B.1.617 by the variant PCR assay were confirmed to be B.1.617.2 and 3/35 were assigned to C.36 lineage with the ONT sequencing. One sample concluded as B.1.351 was assigned to B.1.351.3 lineage by the Pango assessment (S2 Table). Of the 305/332 individual mutations detected by the

variant PCR assay were confirmed by at least one of the sequencing methods. Rest of the unconfirmed spike mutations were masked by N bases in the consensus sequences due to either reduced coverage or low base call accuracy.

## Comparison between illumina and ONT sequencing platforms

**Sequencing statistics.** 37 SARS-CoV-2 samples sequenced using both approaches had more than 45% of the genome coverage and were included in the analysis for comparison of both techniques. Ampliseq SARS-CoV-2 primers used for Illumina library contained 247 amplicons in 2 pools covering >99% of the genome (reference positions between 30 to 29842). The Midnight primer scheme used in the ONT, with just 29 primers in two pools created 1200bp long amplicons and typically covered 54 to 29903 positions. The 150bp Illumina library yielded ~50 million bases, whereas 1200bp ONT library yielded only ~9 million. However, despite the lower number of reads with ONT, it was able to achieve twice the coverage depth of which was achieved by the Illumina platform (Table 1). With the Illumina sequencing technology, the samples required an average of 372,654 reads per sample, whereas ONT achieved even higher coverage depth with an average 9741 reads per sample due to its ability to score ultra-long reads. The higher proportion of ambiguous bases generated by the Illumina platform, resulted in non-detection of several mutations in each sample. We observed one consistent amplicon drop at third amplicon in ONT dataset due to K634N mutation in ORF1a, which is prevalent in Sri Lankan delta sub-lineage, B.1.617.2.AY.28.

Illumina raw reads had an average PHRED quality score (probability of error per base call in a log scale) of 32.35 with a highest score of 36 which is between 1 in 1000 to 1 in 10,000 probability of calling an incorrect base. (99.9–99.99% accuracy). ONT on the other hand had recorded an average score of 10.78 with a 16.2 highest PHRED quality score indicating between 1 in 10 to 1 in 100 probability of error (90% - 99% accuracy). This explains the average

**Table 1. Basic sequencing matrices for Illumina and Oxford Nanopore (ONT) outputs of 37.**

| Sequencing metrics | Illumina | ONT |
|---|---:|---:|
| Number of samples | 37 | 37 |
| Alignment start and end positions | 30–29842 | 54–29903 |
| Mean Coverage depth | 109 | 266 |
| Total Number of reads | 372,654 | 9741 |
| Yielded bases | 50,576,802 | 9,307,884 |
| Fraction of bases aligned | 0.928 | 0.897 |
| Mean Read length | 140 | 945 |
| Average per read identity | 99.6 | 91.4 |
| Average PHRED score | 32.35 | 10.78 |
| No of SNPs | 31 | 36 |
| No of amino acid substitutions | 22 | 25 |
| No of deletions | 14 | 15 |
| No of Amino acid substitutions | 22 | 25 |
| No of frameshift mutations | 0 | 2 |
| % of ambiguous bases | 9% | 6% |
| No of samples with < 10% ambiguous bases | 25 | 29 |
| Successful Pangolin calls | 31 | 34 |
| Successful Scorpio calls | 25 | 31 |
| Run time (h) for 96 samples | 26 | 14 |
| Cost per sample (USD) | ~150–250 | ~10–40 |

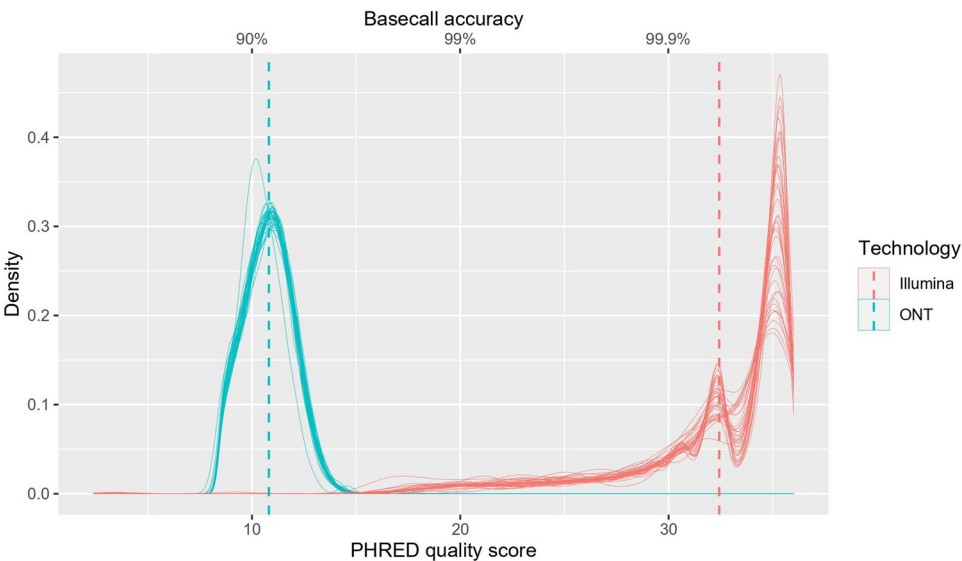

**Fig 1. PHRED base call quality score distribution of samples sequenced by Illumina and ONT.** Distribution plot of PHRED (probability of error per base call in a log scale) quality score (x axis) and error probability (secondary x axis) derived from the PHRED score for the data set sequenced from Illumina (n = 37) and ONT (n = 37). The scores of ONT are shown in blue and Illumina in red. The mean PHRED scores/error probability are shown with the dashed line for each technology. The mean PHRED scores averaged at 32.35 in Illumina reads and 10.78 in ONT.

per read identity to the reference genome of 99.6% in Illumina reads and 91.4% in ONT reads (Table 1 and Fig 1).

**Consensus accuracy.** We used iVar [20] and Medaka workflows to call consensus for Illumina and ONT respectively. ONT detected more average mutations per sample compared to Illumina (36 vs 31) and majority of them were non-synonymous mutations scoring average of 22 and 25 amino acid substitutions in Illumina and ONT respectively. Some known amino acid mutations such as Spike N501Y and NSP12 P323L were constantly missed in Illumina consensus sequences which was caused by masking of those regions due to drop of coverage (Fig 2). However, average number of deletions per sample were between 14 to15 range for

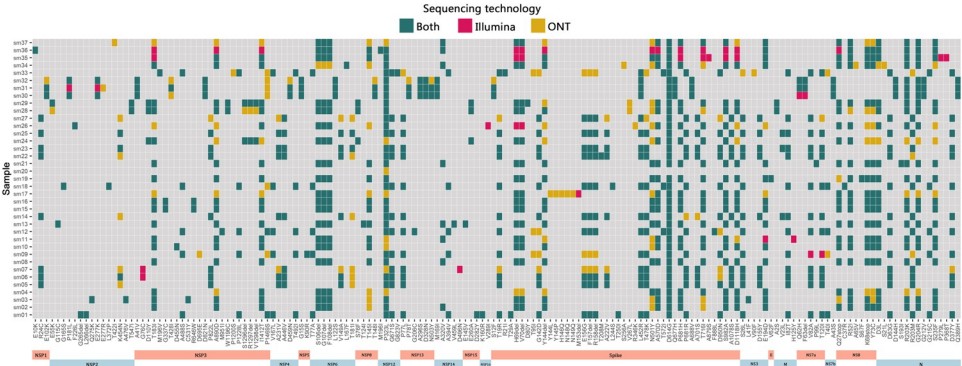

**Fig 2. Comparison of amino acid changes detected in SARS-CoV-2 genomes by both sequencing technologies.** Annotated amino acid substitutions and deletions detected in each sample (n = 37). Mutations colored in green indicates they are synonymously detected by both sequencing technologies, whereas yellow and red indicate mutations detected exclusively by only one technology. The X axis indicates each amino acid, which is denoted by the original amino acid, its position in the protein and the substitution/deletion. Amino acid deletions are denoted by "del".

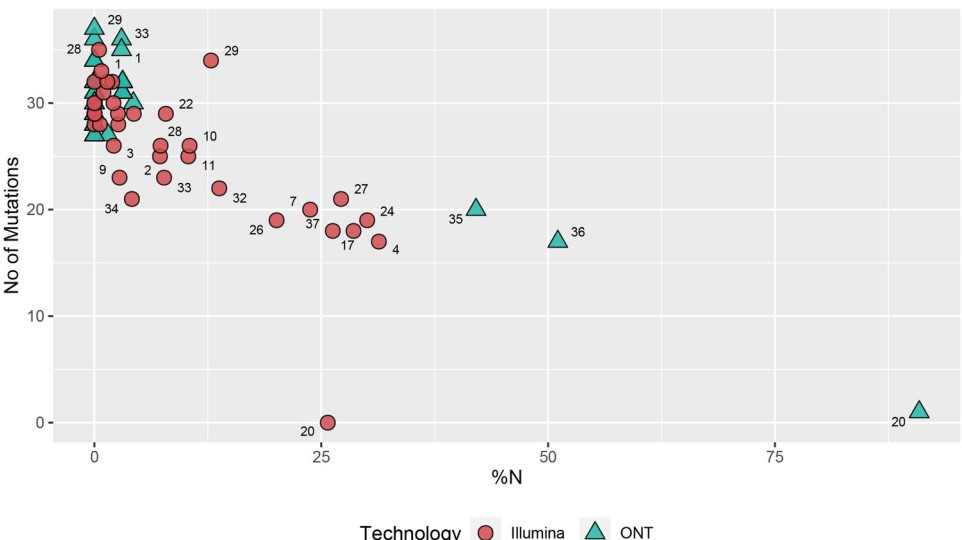

**Fig 3. Percentage of ambiguous bases (% of N) compared to the no of mutations (SNP) detected in each sample.**
The trend between percentage of ambiguous bases is shown in the x axis and number of mutations projected onto the
consensus sequences is shown in the y axis. The numbers displayed on each shape denotes the sample identification
number (n = 37) sequenced by Illumina (red) or ONT (blue). Both technologies had detected a maximum SNPs with
1% - 5% ambiguous bases. The consensus sequences generated by Illumina had a varying percentage of ambiguous
bases between 1% - 30%, whereas ONT sequencing generated either < 5% or more than 40% ambiguous bases due to
its longer read lengths. Altogether ONT had detected more SNPs than Illumina between of 1% - 5% ambiguous bases.

both approaches (Fig 3). Average percentage of ambiguous (N) bases was higher in Illumina
sequences compared to ONT (9% vs 6%), whereas samples with less than 10% ambiguous
bases were higher for ONT compared to Illumina (29 vs 25). The ONT consensus sequences
had 2 frameshift mutations at nucleotide positions of 1634 and 21992, whereas no frameshift
mutations were found in the Illumina dataset.

Due to improved coverage over the SARS-CoV-2 genome, more samples which were
sequenced using the ONT platform were lineage callable by both Pango and Scorpio nomen-
clature (Table 1). Pango lineage calls of 31/37 were identical in both platforms while only 2/37
lineage calls were different between the two sequencing platforms; two ONT datasets classified
as C.36 were called as B.1 and B.1.1.372 with the Illumina datasets. 4/37 samples generated
>31% ambiguous bases with either platform hence failed the lineage calling.

In the combined phylogenetic tree of consensus sequences, 28/37 samples paired with their
counterpart sequences, while those that were not paired together had moderate to high (3% -
31%) ambiguous bases in one of the counterpart sequences. Higher than 98% bootstrap sup-
port can be observed in 21/37 samples which had > 90% genome coverage from both sequenc-
ing technologies (Fig 4).

**Intra-sample variation.** We further analyzed BAM alignments using VarScan2 to deter-
mine the rate of both consensus and sub-consensus variants between two technologies. Collec-
tively, 191 single nucleotide variants (SNVs) and 9 indels were detected in the read pileup of
Illumina dataset, while 226 SNVs and 304 indels were detected in ONT read pileup. However,
175 SNVs were concordant across both datasets suggesting they are true variants. 15 SNVs
detected only in the Illumina dataset are of very low allele frequency (5%-0.1%) whereas 51
SNVs detected only in the ONT dataset are of allele frequency ranging between 52%-0.1% (S3
Table). 8 indels were concordant between both platforms while a significant number (296) of
false positive indels were detected in the ONT dataset with allele frequency between 52% -

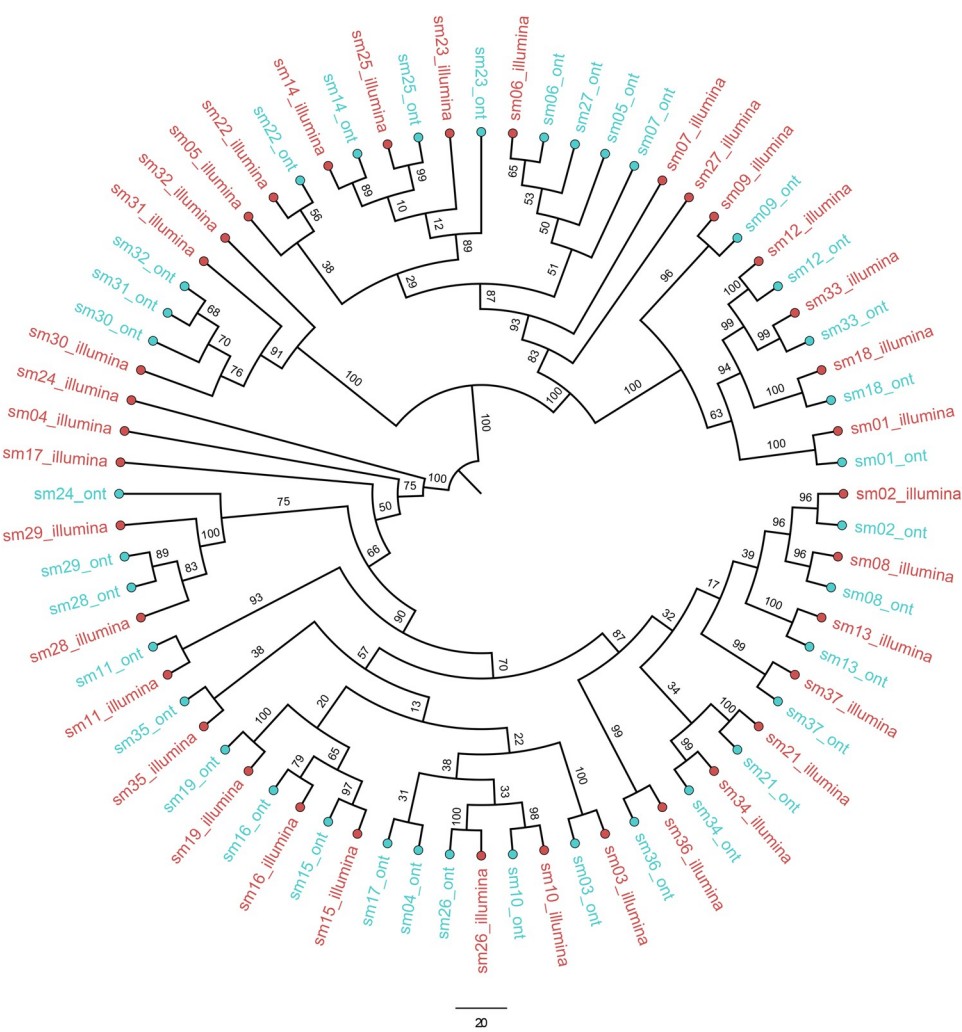

**Fig 4. Combined maximum likelihood phylogenetic tree created using sequence pairs of 37 the samples.** The ML tree was generated using the consensus sequences of each sequencing technology with 1000 bootstrap replicates using TIM2+F+R2 model. Tree is rooted on SARS-CoV-2 reference MN908947.3 and with samples sequences by Illumina coloured red and those sequenced by ONT coloured blue. Bootstrap support values are shown on each branch. 21/37 samples coupled together with < 98% bootstrap support had > 90% genome coverage from both Illumina and ONT datasets while 7/37 samples coupled together with less than 98% bootstrap support. 9/37 of the samples which failed to couple with their counterpart from ONT or Illumina had moderate to high (3% - 31%) ambiguous bases in either sequences.

0.1%. We also observed a clear correlation between the SNV frequencies ($R^2 = 0.79$) while a very weak correlation between indel frequencies ($R^2 = 0.13$) of Illumina and ONT datasets (Fig 5).

## Discussion

In this study we compared the accuracy and sensitivity of a commercial multiple real-time PCR assay for detection of different SNPs with two sequencing technologies in identifying SARS-CoV-2 VOCs. 190 samples which were tested by the variant PCR technique showed 100% concordance with the results of either Illumina or ONT sequencing platforms as far as the VOC assignment. However, 3 samples which were considered to be B.1.617 due to the

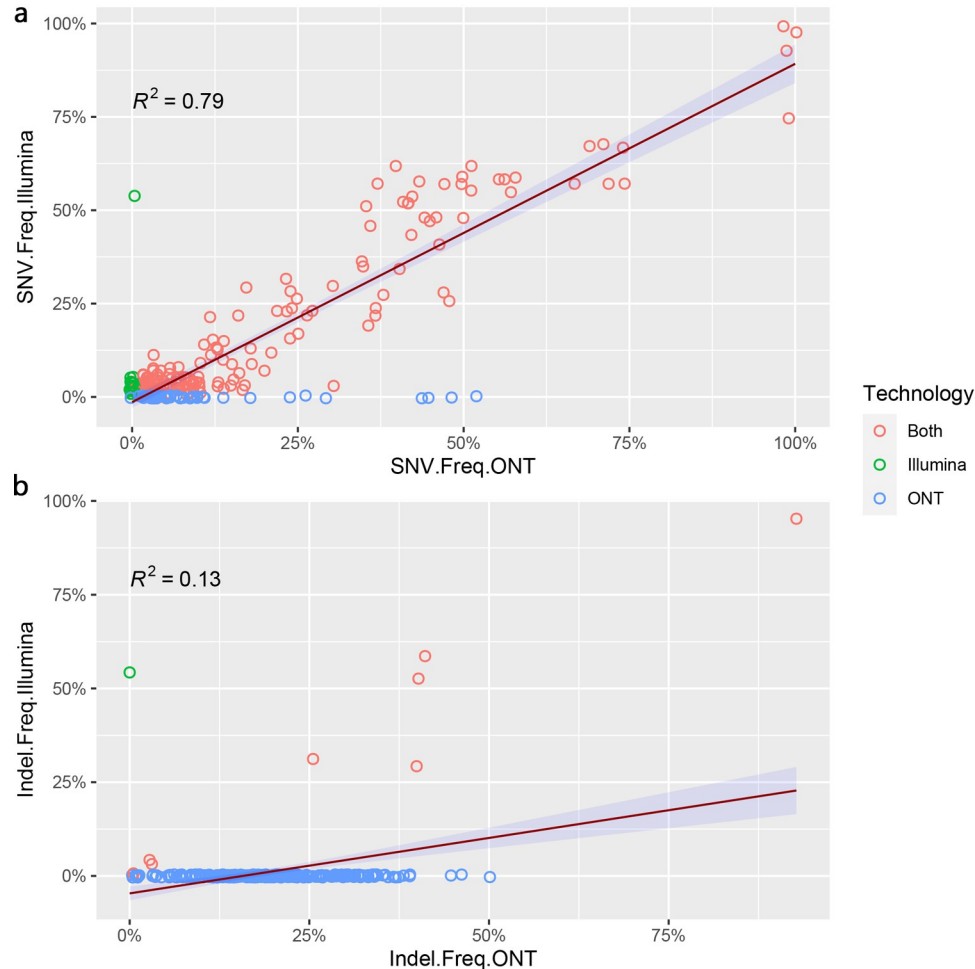

**Fig 5. Correlation of sub-consensus allele frequencies observed for SNV and Indels between two sequencing technologies.** a) Correlation between sub-consensus single nucleotide substitution frequencies observed for Illumina and ONT. Nucleotide substitutions detected exclusively by one technology are indicated in green and blue whereas the substitutions detected by both technologies are colored red. Even though more nucleotide substitutions exclusive to ONT were observed, there is a clear positive correlation ($R^2 = 0.79$) between two sequencing technologies. b) Correlation between sub-consensus indel frequencies observed for Illumina and ONT. More indels exclusive to ONT can be seen with a weak correlation ($R^2 = 0.13$) between the indel frequencies between two technologies suggesting ONT tend to result in more false-positive indels.

presence of the SNP L452R, were assigned to the C.36 Pangolin lineage following sequencing. Since there are many variants with the L452R [21], when multiple SARS-CoV-2 variants are co-circulating, classifying a virus into the B.1.617 lineage based on the L452R mutation alone, would result in inaccuracies. However, for the SNPs E484K, K417T, L452R, K417N, W152C and 69/70 deletion no false positive or false negatives were detected. Therefore, the VOCs assigned to either B.1.1.7 or B.1.351 showed 100% accuracy with both sequencing platforms. Therefore, the variant PCR appears to be a relatively inexpensive and rapid technique to carry out surveillance for SARS-CoV-2 variants in resource poor settings. However, with the dominance of the delta variant globally [22], these variant PCR assays have limited value in detection of new mutations of concern arising in the delta variant that may give rise to higher transmissibility and immune evasion.

In this study we also compared to accuracy and ease of use of two sequencing platforms. The ONT rapid barcoding workflow occupies transposase-based library preparation, which does not require individual sample washes and allows samples to be processed uniformly without quantification or normalization [23]. For Illumina, traditional ligation-based library preparation was used which required extended preparation time and effort. Run time per 96 samples on ONT is nearly a half the time required for Illumina (26 hours compared to 14 hours) mainly due to the ability of real-time data analysis with ONT. The rapid barcoding library preparation method used for the ONT platform also required less reagents and therefore, was cheaper than Illumina sequencing. Although the single ended barcoding of the transposase-based libraries is thought to result in improper demultiplexing and sample crossover, it has shown to rarely affect variant calling and consensus generation in ONT [12].

The Illumina sequencing produced ~200bp long amplicons, whereas the ONT sequencing platform produced 1200bp. The long amplicons generated by the ONT sequencing achieved higher coverage and less ambiguous bases even with 10,000 reads (after filtering). As a result of this, certain mutations were not detected by the Illumina platform due to a larger proportion of ambiguous bases. For instance, 34/37 of the samples sequenced by ONT had <5% ambiguous bases, while 21/37 samples sequenced using Illumina generated <5%. However, the mean PHRED scores averaged at 32.35 in Illumina reads but 10.78 in ONT. This difference results in a base call error probability of 1 in 10 by the ONT and 1 in 1000 for Illumina sequencing platform. The reduced accuracy in base calls and increase the frequency of erroneous bases can be minimized by using high accuracy or super accuracy models of guppy basecaller [24]. 31/37 samples had identical Pango lineage calls despite having ambiguous bases as high as 30% in the Illumina dataset. Only two samples (13.8% - 0.02% ambiguous bases) resulted in different lineage calls, which were non-VOCs (C.36 and B.1.1.372). Even though it is evident that consensus sequences with >30% ambiguous bases fail to be assigned to a Pango pipeline, the two platforms resulted in 100% concordance when it comes to calling VOCs from the sequences with >70% genome coverage. While short read sequencing is considered as the gold standard for sequencing of viral genomes [25], despite ONT having a slightly higher error rate, ONT appears to generate high quality data at a very affordable cost. Therefore, ONT appears to be the most cost effective, high throughout sequencing technology, especially suited for countries with limited resources for genomic surveillance and for identification of emerging variants of concern.

In summary, we have compared the usefulness, accuracy and reliability of two sequencing technologies and also compared the usefulness of a commercial multiplex real-time PCR for surveillance of VOCs, by identification of SNPs associated with the VOCs. We found that the multiplex real-time PCR assay detected the alpha variant with 96.3% accuracy and the delta variant with 85% accuracy, when compared with Illumina sequencing technology. Although the ONT had a slightly higher error rate compared to the Illumina technology, it achieved higher coverage with a lower number or reads, generated less ambiguous bases and was significantly less expensive than Illumina sequencing technology.

## Supporting information

**S1 Table. Metadata of 190 samples used in the comparison.**
(XLSX)

**S2 Table. Comparison statistics of RT-PCR vs each sequencing technology.**
(XLSX)

**S3 Table. Sub-consensus SNV and Indel frequency data of ONT and Illumina.**
(XLSX)

**S1 Methods. Methods in detail used in the experiment.**
(RTF)

## Author Contributions

**Conceptualization:** Diyanath Ranasinghe, Chandima Jeewandara, Gathsaurie Neelika Malavige.

**Data curation:** Osanda Dissanayake, Dinuka Guruge.

**Formal analysis:** Diyanath Ranasinghe, Tibutius Thanesh Pramanayagam Jayadas, Deshni Jayathilaka.

**Funding acquisition:** Chandima Jeewandara, Gathsaurie Neelika Malavige.

**Investigation:** Diyanath Ranasinghe, Tibutius Thanesh Pramanayagam Jayadas, Deshni Jayathilaka, Osanda Dissanayake, Dinuka Ariyaratne, Dumni Gunasinghe, Laksiri Gomes, Ayesha Wijesinghe.

**Methodology:** Dinuka Ariyaratne, Dumni Gunasinghe.

**Project administration:** Chandima Jeewandara, Dinuka Guruge, Ruwan Wijayamuni.

**Resources:** Chandima Jeewandara, Dinuka Guruge, Ruwan Wijayamuni.

**Writing – original draft:** Diyanath Ranasinghe, Tibutius Thanesh Pramanayagam Jayadas, Gathsaurie Neelika Malavige.

**Writing – review & editing:** Gathsaurie Neelika Malavige.

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
