## [Decision Letter · Decision Letter 0]

10 Jan 2022

PONE-D-21-37979Comparison of different sequencing techniques with multiplex real-time PCR for detection to identify SARS-CoV-2 variants of concernPLOS ONE

Dear Dr. Malavige,

Thank you for submitting your manuscript to PLOS ONE. After careful consideration, we feel that it has merit but does not fully meet PLOS ONE’s publication criteria as it currently stands. Therefore, we invite you to submit a revised version of the manuscript that addresses the points raised during the review process.

We look forward to receiving your revised manuscript.

Kind regards,

Ruslan Kalendar

Academic Editor

PLOS ONE

Journal Requirements:

Reviewers' comments:

Reviewer's Responses to Questions

**Comments to the Author**

1. Is the manuscript technically sound, and do the data support the conclusions?

Reviewer #1: Yes

2. Has the statistical analysis been performed appropriately and rigorously? 

Reviewer #1: No

3. Have the authors made all data underlying the findings in their manuscript fully available?

Reviewer #1: Yes

4. Is the manuscript presented in an intelligible fashion and written in standard English?

Reviewer #1: Yes

5. Review Comments to the Author

Reviewer #1: 

In this study, RTPCR for single nucleotide polymorphisms was high accuracy for VOC identification. There were three false positive results for B.1.617 due to an isolated L452R mutation but B.1.1.7 or B.1.351 calls were 100% accurate. The authors also compared Oxford Nanopore technology (ONT) to Illumina sequencing. Higher call accuracy was observed with Illumina among 37 samples consistent with what would be expected but ONT had higher sequence coverage. This study was conducted prior to emergence of the omicron VOC and further experiments would need to be performed to evaluate these RTPCRs for differentiation from omicron. Overall, this was a proof-of-concept study that RTPCR may be useful for differentiating known predominant VOCs and ONT is acceptable in resource-limited settings. This is a relevant topic but there were major methodological limitations.

Major limitations: While ONT and Illumina Nextseq 550 platforms are compared, they are then used interchangeably for the RTPCR comparison. The results should be separated for comparisons to each gold standards. The immediate translation of RTPCR findings is limited by the shift to the omicron variant, but there is still utility in this study as a proof-of-concept.

If the accuracy was unchanged between sequencing platforms, this needs to be described. This is particularly relevant when 7 of 27 lineage calls were different by different methods. A table for information about the RTPCR performance compared to each sequencing method would be helpful to this end. The authors should consider including a 2x2 table for sensitivity/specificity in the supplement or describe it more clearly (i.e., the % agreement) in the results to support statements of good sensitivity and accuracy.

The dates or months that samples were collected would be useful to put into context each sequencing method for the RTPCR comparison. The dates/months that variants were introduced in this population should also be described.

Minor:

The title "Comparison of different sequencing techniques with multiplex real-time PCR for detection to identify SARS-CoV-2 variants of concern" is not grammatically correct. It should be "for detection" or "for identification"

In the abstract, regarding the sentence “34/37 of the samples sequenced by ONT had <5% ambiguous bases, while 21/37 samples sequenced using the Illumina generated <15% ambiguous bases”, it is not clear why cutoffs change. The authors could use the sentence from the results “samples with less than 10% ambiguous bases were higher for ONT compared to 21 Illumina (29 vs 25).”

Only two RTPCR panels described. In abstract and the introduction, the authors should state two panels by the same commercial vendor are used. Currently, it seems like many panels were going to be compared.

Under methods, it is unclear what kit, primers or probes were used for initial detection for the 30 cycle threshold cut off. It should be described if RTPCR was run in duplicate or triplicate.

“Due to improved coverage over the SARS-CoV-2 genome, 1 more samples which were sequenced using the ONT platform were lineage call-able by both Pango and Scorpio nomenclature.” – Please list the number in the text and/or cite the table.

In the discussion, I would recommend writing “less expensive” rather than “cheaper” as a potential benefit. Cost effectiveness was not evaluated. So, I would consider reframing the language to state that the error rate is acceptable when resources for Illumina sequencing may not be available.

6. PLOS authors have the option to publish the peer review history of their article (what does this mean?). If published, this will include your full peer review and any attached files.

Reviewer #1: No

---

## [Author Response · Author response to Decision Letter 0]

12 Feb 2022

7th February 2022

Dr. Ruslan Kalendar

Academic Editor

PLOS ONE

Dear Dr. Kalendar,

Submission of manuscript titled ‘Comparison of different sequencing techniques for identification of SARS-CoV-2 variants of concern with multiplex real-time PCR’

We wish to thank the reviewers for carefully going through our manuscript and the useful suggestions they have made. We have incorporated all these changes in the revised version and letter and have addressed all issues raised by the reviewers. We have highlighted the changes in the revised manuscript.

Reviewer #1: 

In this study, RTPCR for single nucleotide polymorphisms was high accuracy for VOC identification. There were three false positive results for B.1.617 due to an isolated L452R mutation but B.1.1.7 or B.1.351 calls were 100% accurate. The authors also compared Oxford Nanopore technology (ONT) to Illumina sequencing. Higher call accuracy was observed with Illumina among 37 samples consistent with what would be expected but ONT had higher sequence coverage. This study was conducted prior to emergence of the omicron VOC and further experiments would need to be performed to evaluate these RTPCRs for differentiation from omicron. Overall, this was a proof-of-concept study that RTPCR may be useful for differentiating known predominant VOCs and ONT is acceptable in resource-limited settings. This is a relevant topic but there were major methodological limitations.

Major limitations: 

1) While ONT and Illumina Nextseq 550 platforms are compared, they are then used interchangeably for the RTPCR comparison. The results should be separated for comparisons to each gold standards. The immediate translation of RTPCR findings is limited by the shift to the omicron variant, but there is still utility in this study as a proof-of-concept.

Response: We highly value and appreciate the reviewer’s suggestions. We have separated the RT-PCR vs NGS comparison into two sections calculating percentages for each NGS technology, in the revised version of the manuscript.

We also completely agree with the comment the reviewer has made regarding detection of Omicron. Since the spread of the Omicron variant in Sri Lanka, we are in the process of assessing our in-house RTPCR assay for identifying Omicron and we have been able to successfully differente Omicron BA.1 from other variants (by presence of E484K, N501Y, HV69/70, L452R, and K417N). We have also been able to identify BA.2 from BA.1 due to the absence of HV69.70 in BA.2. However, as we have just started assessing the usefulness of this RTPCR for BA.1 and BA.2, we have not included this data in this manuscript.

2) If the accuracy was unchanged between sequencing platforms, this needs to be described. This is particularly relevant when 7 of 27 lineage calls were different by different methods. 

Response: We thank the reviewer for this suggestion. This section was rewritten taking genome coverage into account and ignoring the differences in sub lineages (AY classification was changed by Pango team after we submitted the paper so it can be confusing to use those obsolete sub lineages). Only a classification at VOC level was used (pango lineage version 2021-08-09)

3) A table for information about the RTPCR performance compared to each sequencing method would be helpful to this end. The authors should consider including a 2x2 table for sensitivity/specificity in the supplement or describe it more clearly (i.e., the % agreement) in the results to support statements of good sensitivity and accuracy.

Response: we thank the reviewer for this suggestion. We have added summary table and a Venn diagram to supplementary table 2.

4) The dates or months that samples were collected would be useful to put into context each sequencing method for the RTPCR comparison. The dates/months that variants were introduced in this population should also be described.

Response: We thank the reviewer for this suggestion. We have included the collection dates and deposited dates for all the samples in supplementary table 1. In addition, we have added a few lines to the methods section regarding the outbreak periods.

Minor:

5) The title "Comparison of different sequencing techniques with multiplex real-time PCR for detection to identify SARS-CoV-2 variants of concern" is not grammatically correct. It should be "for detection" or "for identification"

Response: We apologize for this mistake and we have corrected it as suggested by the reviewer. 

6) In the abstract, regarding the sentence “34/37 of the samples sequenced by ONT had <5% ambiguous bases, while 21/37 samples sequenced using the Illumina generated <15% ambiguous bases”, it is not clear why cutoffs change. The authors could use the sentence from the results “samples with less than 10% ambiguous bases were higher for ONT compared to 21 Illumina (29 vs 25).”

Response: We thank the reviewer for pointing this out. We have corrected this throughout the text.

7) Only two RTPCR panels described. In abstract and the introduction, the authors should state two panels by the same commercial vendor are used. Currently, it seems like many panels were going to be compared.

Response: We apologize for this mistake. We have corrected this in the revised version of the manuscript. 

8) Under methods, it is unclear what kit, primers or probes were used for initial detection for the 30-cycle threshold cut off. It should be described if RTPCR was run in duplicate or triplicate.

Response: We apologize for this omission. We have included this in the methods section in the revised version. .

9) “Due to improved coverage over the SARS-CoV-2 genome, 1 more sample which were sequenced using the ONT platform were lineage call-able by both Pango and Scorpio nomenclature.” – Please list the number in the text and/or cite the table.

Response: we thank the reviewer for pointing out this omission. We have corrected this in the revised version. 

10) In the discussion, I would recommend writing “less expensive” rather than “cheaper” as a potential benefit. Cost effectiveness was not evaluated. So, I would consider reframing the language to state that the error rate is acceptable when resources for Illumina sequencing may not be available.

Response: We agree with the reviewer that this is a more suitable word to use. We have corrected this in the revised version of the manuscript. 

Thank you for considering our manuscript for publication with PLOS One.

Yours Sincerely,

Prof. Neelika Malavige

---

## [Editor Report · Decision Letter 1]

28 Feb 2022

Comparison of different sequencing techniques for identification of SARS-CoV-2 variants of concern with multiplex real-time PCR

PONE-D-21-37979R1

Dear Dr. Malavige,

We’re pleased to inform you that your manuscript has been judged scientifically suitable for publication and will be formally accepted for publication once it meets all outstanding technical requirements.

Kind regards,

Ruslan Kalendar

Academic Editor

PLOS ONE

---

## [Editor Report · Acceptance letter]

15 Mar 2022

PONE-D-21-37979R1 

Comparison of different sequencing techniques for identification of SARS-CoV-2 variants of concern with multiplex real-time PCR 

Dear Dr. Malavige:

I'm pleased to inform you that your manuscript has been deemed suitable for publication in PLOS ONE. Congratulations! Your manuscript is now with our production department. 

Kind regards, 

on behalf of

Professor Ruslan Kalendar 

Academic Editor

PLOS ONE